# NETosis and Nucleosome Biomarkers in Septic Shock and Critical COVID-19 Patients: An Observational Study

**DOI:** 10.3390/biom12081038

**Published:** 2022-07-27

**Authors:** Laure Morimont, Mélanie Dechamps, Clara David, Céline Bouvy, Constant Gillot, Hélène Haguet, Julien Favresse, Lorian Ronvaux, Julie Candiracci, Marielle Herzog, Pierre-François Laterre, Julien De Poortere, Sandrine Horman, Christophe Beauloye, Jonathan Douxfils

**Affiliations:** 1Research and Development Department, Qualiblood s.a., 5000 Namur, Belgium; laure.morimont@qualiblood.eu (L.M.); clara.david@qualiblood.eu (C.D.); celine.bouvy@qualiblood.eu (C.B.); 2Namur Thrombosis and Hemostasis Center, Namur Research Institute for Life Sciences, Department of Pharmacy, University of Namur, 5000 Namur, Belgium; constant.gillot@unamur.be (C.G.); helene.haguet@unamur.be (H.H.); julien.favresse@slbo.be (J.F.); lorian.ronvaux@unamur.be (L.R.); 3Cardiovascular Intensive Care, Cliniques Universitaires St Luc, 1200 Brussels, Belgium; melanie.dechamps@saintluc.uclouvain.be (M.D.); pierre-francois.laterre@uclouvain.be (P.-F.L.); 4Pôle de Recherche Cardiovasculaire, Institut de Recherche Expérimentale et Clinique (IREC), Université Catholique de Louvain, 1200 Brussels, Belgium; julien.depoortere@uclouvain.be (J.D.P.); sandrine.horman@uclouvain.be (S.H.); christophe.beauloye@uclouvain.be (C.B.); 5Department of Laboratory Medicine, Clinique St Luc Bouge, 5004 Namur, Belgium; 6Belgian Volition SRL, Parc Scientifique Crealys, 5032 Isnes, Belgium; j.candiracci@volition.com (J.C.); m.herzog@volition.com (M.H.); 7Division of Cardiology, Cliniques Universitaires St Luc, 1200 Brussels, Belgium

**Keywords:** COVID-19, septic shock, SARS-CoV-2, NETs’ formation, nucleosomes

## Abstract

**Background**: Neutrophil extracellular traps’ (NETs’) formation is a mechanism of defense that neutrophils deploy as an alternative to phagocytosis, to constrain the spread of microorganisms. **Aim**: The aim was to evaluate biomarkers of NETs’ formation in a patient cohort admitted to intensive care unit (ICU) due to infection. **Methods**: Forty-six septic shock patients, 22 critical COVID-19 patients and 48 matched control subjects were recruited. Intact nucleosomes containing histone 3.1 (Nu.H3.1), or citrullinated histone H3R8 (Nu.Cit-H3R8), free citrullinated histone (Cit-H3), neutrophil elastase (NE) and myeloperoxidase (MPO) were measured. **Results**: Significant differences in Nu.H3.1 and NE levels were observed between septic shock and critical COVID-19 subjects as well as with controls (*p*-values < 0.05). The normalization of nucleosome levels according to the neutrophil count improved the discrimination between septic shock and critical COVID-19 patients. The ratio of Nu.Cit-H3R8 to Nu.H3.1 allowed the determination of nucleosome citrullination degree, presumably by PAD4. **Conclusions**: H3.1 and Cit-H3R8 nucleosomes appear to be interesting markers of global cell death and neutrophil activation when combined. Nu.H3.1 permits the evaluation of disease severity and differs between septic shock and critical COVID-19 patients, reflecting two distinct potential pathological processes in these conditions.

## 1. Introduction

Neutrophil extracellular traps’ (NETs’) formation is an innate immune response that neutrophils deploy in addition to phagocytosis, to constrain the spread of fungi, large bacteria, viruses and several other microorganisms [1]. Given that the neutrophil arsenal can also damage host tissues, its deployment is tightly regulated [2]. Neutrophil extracellular traps are large, extracellular, web-like structures composed of cytosolic and granule proteins that are assembled on a scaffold of decondensed chromatin [1]. Markers of this physiological phenomenon include nucleosomes and histones and their epigenetic modifications, neutrophil elastase (NE), myeloperoxidase (MPO), calprotectin, cathelicidins, defensins and actin, among others [3]. Interestingly, it has been demonstrated that the composition of NETs varies depending on the stimulus [4].

The nucleosome is the fundamental unit of chromatin, which is formed of 147 base pairs of DNA wrapped around a histone core: H2A, H2B, H3 and H4. The majority of cell-free DNA (cfDNA) are stabilized within nucleosomes and the majority of linker DNA are rapidly degraded to leave mono and short poly nucleosomes via apoptotic and necrotic pathways [5]. It has been shown that nucleosomes can be actively released into the circulation from dead cells as a result of the activity of factor VII-activating proteases [6]. Upstream of the nucleosomes’ release into the circulation in the NET formation process, MPO is stimulated by the generation of reactive oxygen species, which are produced by NADPH oxidase. MPO triggers the activation and translocation of NE from azurophilic granules to the nucleus where NE proteolytically processes nucleosomal histones to promote extensive chromatin decondensation. MPO also binds chromatin and synergizes with NE in decondensing chromatin [7]. Another chromatin modification implicated in chromatin decondensation is histone deamination (or citrullination), which is driven by protein arginine deiminase 4 (PAD4), a nuclear enzyme that citrullinates arginine residues, converting amine groups into ketones [8]. Citrullination by PAD4 has a duplicitous nature since it may stimulate NET formation, while, on the other hand, it may inhibit macrophage oxidative burst and reduce the antimicrobial activities of neutrophils, for example, by making histone H3.1 more prone to being degraded by NE [9]. Such post-translational modifications of histones are essential in the controlled balance of NETs formation between host defense and host damage.

A recent report also showed that cell cfDNA and NET markers such as citrullinated histone H3 and MPO-DNA complexes were elevated in COVID-19 [10]. In addition to these markers, nucleosomes have also been proposed as potential biomarkers for NET formation in plasma [11] and to monitor COVID-19 progression [12]. An in vitro study described that nucleosomes, derived from the micrococcal nuclease (MNase) digestion of NET linker DNA, induce the expression of cytokine IL-1β from monocytes in vitro and citrullination enhances this pro-inflammatory signaling through enhanced binding to TLR4, promoting NET inflammation [13]. Septic shock is a severe clinical syndrome defined as the host body’s dysregulated response to infection leading to a life-threatening organ dysfunction. It is generally described as being the result of a bacterial infection, but virus, fungal or other pathogens may also be responsible for this clinical condition [14,15]. COVID-19 has been suggested to be a typical viral septic pulmonary infection, which causes systemic inflammation and dysregulation of the immune response leading to multiple organ dysfunction and death [16,17].

We previously reported the results of a study comparing clinical outcomes, inflammatory reaction and coagulopathy between critical COVID-19 and septic shock patients [18]. This study revealed that critical COVID-19 patients differed from those with septic shock at admission into the intensive care unit (ICU). Namely, COVID-19 patients had higher levels of IL-1β and T lymphocyte activation (including IL-7), whereas septic shock displayed higher levels of IL-6 and IL-8, and a more significant myeloid response (including triggering receptors expressed on myeloid cells-1 (TREM-1 and IL-1ra). In addition, markers of coagulopathy also differed as highlighted by higher levels of soluble tissue factor and fibrinogen, less platelet and antithrombin consumption, and fewer fibrinolysis alterations in critical COVID-19 compared to septic shock patients [18]. Based on these results, we suggested potential therapeutic strategies, in particular, recombinant IL-1ra and recombinant tissue factor pathway inhibitor (TFPI), in order to modulate these two overstimulated pathways in COVID-19 patients [18]. Interestingly, both conditions have been linked to excessive NET formation [19,20], but the direct comparison of NETs’ formation biomarkers was not part of our initial investigations.

Therefore, the aim of this study was to evaluate circulating nucleosomes and neutrophil activation markers in these two patient populations. Viral infections encompass a broad spectrum of pathogens and diseases in humans but—apart from specific clinical situations such as epidemics/pandemics—are rarely the primary cause of sepsis. In a recent large international point prevalence study, viruses were documented in less than 4% of infections [21]. Historically, influenza has been one of the more common viral causes of sepsis. However, it is unclear to what extent the primary viral infection as opposed to bacterial pneumonia co-infection is the cause of organ dysfunctions in these patients. Nevertheless, SARS-CoV-2, causing COVID-19, is now responsible for many cases of infection and sepsis, and this is also the reason why we decided to investigate these two populations.

## 2. Materials and Methods

### 2.1. Population and Clinical Outcome

The population has already been described previously in detail and is summarized in Appendix A [18]. Briefly, patients with critical COVID-19 who were admitted to the ICU for moderate or severe acute respiratory distress syndrome (ARDS) due to SARS-CoV-2 infection were included within five days of admission. ARDS was diagnosed according to the Berlin definition [22], and SARS-CoV-2 infection was demonstrated by real-time reverse transcription PCR on nasopharyngeal swabs. Septic shock was defined according to the Sepsis-3 definition as sepsis with vasopressor therapy needed to elevate the mean arterial pressure ≥ 65 mmHg and lactate levels > 2 mmol/L despite adequate fluid resuscitation of 30 mL/kg of intravenous crystalloids within 6 h [15]. Patients with septic shock admitted to the ICU were included within two days of admission. Control patients with matched age, gender and comorbidities were recruited at a central laboratory consultation. Similar exclusion criteria for inclusion were applied to all groups and included therapeutic anticoagulation (oral or parenteral, including heparin, fondaparinux, vitamin K antagonists, and direct oral anticoagulants), recent (within less than one month) chemotherapy, active inflammatory disease, hemophilia and other coagulopathies, previous history of thrombocytopenia (<100,000 platelets/mm^3^), cirrhosis (Child–Pugh > A), recent (within less than 48 h) major surgery (infection source control for septic shock patients), cardiac arrest during ICU stay and decision of care limitations. All septic and COVID-19 patients received thromboprophylaxis using low-molecular-weight heparin (LMWH; nadroparin 3800 IU/days subcutaneously). The demographic characteristics and past medical history were similar among the three groups, except that the COVID-19 group included fewer smokers and oncologic patients. Sampling was performed at least 6 h after LMWH injection. Among patients with COVID-19, those on antibiotics for any suspected or confirmed bacterial confections were formally excluded. Patient prognosis was assessed using acute physiologic assessment and chronic health evaluation II (APACHE-II) and sequential organ failure assessment (SOFA) scores [23,24]. The ethics committee approved the study protocol, and all patients signed their informed consent (B403201938590, NCT04107402). A protocol amendment was made to include COVID-19 patients in the ongoing study on septic shock patients. This amendment did not require a matching between COVID-19 patients and septic shock, explaining why some differences can be observed in the clinical characteristics of this subpopulation compared to controls and septic shock patients. All authors had full access to primary clinical data.

### 2.2. Blood Sample Collection

Blood samples were collected through the central venous catheter in all ICU patients and by venous punctures in the control group. Venous blood was collected using vacutainer tubes containing CPDA. After two centrifugation runs at >1500 *g* for 15 min enabling platelet isolation, plasma was collected, divided into 1 mL aliquots and stored at −80 °C until analysis. Frozen plasma samples were thawed in a water bath at 37 °C for maximum 10 min and mixed gently just before experiments. All tests were performed within 4 h of thawing.

### 2.3. Circulating Nucleosomes, Neutrophil Activation and Inflammatory Biomarkers

Nucleosomes containing histone H3.1 or containing citrullinated nucleosome histone H3R8 were measured using the Nu.Q^®^ H3.1 and Nu.Q^®^ H3R8Cit ELISA assays from Volition (Belgian Volition, Isnes, Namur, Belgium). These assays use anti-histone H3.1 or an anti-citrullinated histone H3R8 as capture antibodies with an anti-nucleosome detection antibody to ensure only histones within intact nucleosomes are quantified. Details on analytical performance of the Nu.Q H3.1 assay can be found in the Instruction for Use [25]. The Nu.Q H3R8Cit is currently for research use only and no detail on analytical performances is provided by the manufacturer. Free citrullinated histone H3 (Cit-H3) (citrullinated at R2, R8 and R17) were measured using the Cayman citrullinated histone H3 ELISA kit (Cayman Chemical, Ann Arbor, MI, USA). Neutrophil elastase and MPO were measured using the Human Neutrophil Elastase/ELA2 DuoSet ELISA and the Human Myeloperoxidase Quantikine ELISA Kit (R&D systems, Minneapolis, MN, USA). Cytokines and chemokines were measured using the Bio-Plex Pro Human Cytokine 27-plex Assay, and ICAM-1 and VCAM-1 were measured by mixing Bio-Plex Pro Human cytokines ICAM-1 and VCAM-1 sets (ICAM-VCAM). Both were measured on a Bio-Plex 200 (Bio-Rad Laboratories N.V., Temse, East Flanders, Belgium). A complete list of investigated biomarkers is summarized in Appendix A. All tests were performed according to the manufacturer’s recommendations.

### 2.4. Statistical Analyses

The statistical analyses were performed using GraphPad Prism (GraphPad Prism 9.3.1 for macOs, GraphPad Software, San Diego, CA, USA). Descriptive statistics were used, and results were reported as median and 10th–90th percentile. Data were subjected to Kolmogorov–Smirnov normality test and standard deviations between groups were assessed by Brown–Forsythe tests. If data were not normally distributed, log transformations were applied when appropriate. The categorical variables were analyzed using the Chi-squared test. Differences between groups, i.e., controls, septic shock and critical COVID-19, for all parameters were assessed using an ordinary two-way ANOVA with uncorrected Fischer’s significant difference multiple comparison on log-transformed data. For comparison between the 3 groups, multiple comparisons were not corrected because it was assumed in the design that controls are different from septic shock and critical COVID-19 patients so there was no need to correct the comparison for this group [26]. Septic shock and critical COVID-19 groups were then stratified according to their APACHE-II and SOFA status, and NETs’ formation biomarkers results were compared between these different subgroups, i.e., 3 groups for APACHE-II score and 4 groups for SOFA score, using ordinary two-way ANOVA with *p*-value corrected for multiple comparisons using a Tukey’s multiple comparison test. For APACHE-II, stratification was performed for scores of 0 to 15, 16 to 25 and 26 to 35. For SOFA, stratification was performed for scores of 0 to 6, 7 to 9, 10 to 12 and ≥13. Comparison of NETs’ formation markers between survivors and non-survivor subjects was conducted in the septic shock and critical COVID-19 cohorts only and results were compared using an unpaired *t*-test. All *p*-values were set as significant at *p* ≤ 0.05 and corrected when appropriate. Pearson’s correlation matrixes were also performed to test the correlation between all parameters. Individual Pearson’s r above 0.70 were considered strong correlations.

## 3. Results

Data on baseline characteristics and clinical outcomes of critical COVID-19 and septic shock patients and data on cytokines and hemostasis parameters have previously been reported in part [18] and are summarized in Appendix A.

### Circulating Nucleosomes and Neutrophil Activation Biomarkers

Circulating nucleosomes and neutrophil activation markers include H3.1 nucleosomes (Nu.H3.1), citrullinated H3R8-nucleosomes (Nu.Cit-H3R8), free citrullinated histone (Cit-H3), NE and MPO. As these markers are or may be related to the activation of neutrophils, a normalization of these parameters by the counting of neutrophils at individual levels has been conducted (Figure 1) as well as the ratios of Cit-H3/Nu.H3.1, Cit-H3/Nu.Cit-H3R8 and Nu. H3R8/Nu.H3.1 (Figure 2).

Overall, all biomarkers investigated in this study were different in control subjects versus critical COVID-19 or septic shock subjects. Nu.H3.1 was higher in critical COVID-19 patients compared to septic shock patients (median [10th–90th percentile]: 2533 [706–4389] ng/mL versus 862 [252–9398] ng/mL, *p* = 0.0020). NE was higher in septic shock patients compared to critical COVID-19 patients (median [10th–90th percentile]: 102 [41.8–478] ng/mL versus 57.2 [20.4–178] ng/mL, *p* = 0.0002). The other NETs formation biomarkers investigated in this study were not statistically different between critical COVID-19 and septic shock patients (Appendix A).

The neutrophil count was higher in septic shock patients compared to critical COVID-19 patients (median [10th–90th percentile]: 14.2 [5.11–29.9] × 10^3^/µL versus 7.61 [4–12.3] × 10^3^/µL, *p* = 0.0025) (Appendix A). Normalization by the neutrophil count gave better discrimination between critical COVID-19 and septic shock patients for Nu.H3.1 but not for NE (Figure 1). Indeed, the normalization of Nu.H3.1 by the neutrophil count leads to a more significant difference between COVID-19 and septic shock patients, while a difference no longer appears between these two populations when normalizing NE to neutrophil count.

The ratios of Cit-H3/Nu.H3.1 and Nu.Cit-H3R8/Nu.H3.1 were statistically different between COVID-19 and sepsis patients (median [10th–90th percentile]: 3.62 × 10^−3^ [6.10 × 10^−4^–8.43 × 10^−3^] versus 1.16 × 10^−3^ [3.18 × 10^−4^–4.59 × 10^−3^], *p* = 0.0372 and 0.073 [0.018–0.16] versus 0.031 [0.011–0.081], *p* < 0.0001 for Cit-H3/Nu.H3.1 and Nu.Cit-H3R8/Nu.H3.1, respectively), while the ratio of Cit-H3/Nu.Cit-H3R8 did not show difference (median [10th–90th percentile]: 0.045 [0.020–0.104] versus 0.047 [0.019–0.109], *p* = 0.9769) (Figure 2). In addition, no statistical differences were observed for these markers between survivors and non-survivors (*p* > 0.05) (Table 1, Figure 3 and Figure 4, non-transparent symbols).

Stratification of the two disease cohorts according to the APACHE-II and the SOFA scores are reported in Figure 3 and Figure 4, respectively. The levels of Nu.H3.1 increased with higher APACHE-II and SOFA scores in septic shock patients, while in critical COVID-19 patients, the SOFA score did not correlate with Nu.H3.1 levels. An inverse correlation was observed for the APACHE-II scores and levels of Nu.H3.1 in COVID-19 patients (Appendix A). MPO levels were also different between SOFA 0–6 and SOFA ≥ 13 (median [10th–90th percentile]: 187.7 [53.9–665.2] ng/mL versus 2269.2 [229.7–5540.1] ng/mL, *p* = 0.0193) and SOFA 7–9 and SOFA ≥ 13 (median [10th–90th percentile]: 217.0 [76.0–4365.3] ng/mL versus 2269.2 [229.7–5540.1], *p* = 0.0332) in septic shock patients (Figure 4). None of the other measured parameters showed differences in septic shock and critical COVID-19 subgroups. Ratios of Nu.H3.1/neutrophils and Nu.Cit-H3R8/Nu.H3.1 did not change the significance of the stratification between the different scores within a group, i.e., septic shock or critical COVID-19, but for similar APACHE-II or SOFA scores, Nu.H3.1 levels, Nu.H3.1/neutrophils and Nu.Cit-H3R8/Nu.H3.1 were statistically different between septic shock and critical COVID-19 patients (Table 2). Nu.H3.1 levels and the ratio of Nu.H3.1/neutrophils were superior in critical COVID-19 patients versus septic shock patients, while the ratio of Nu.Cit-H3R8/Nu.H3.1 was lower in COVID-19 patients (Figure 4 and Table 2).

There was a stronger correlation, as defined by a Pearson r above 0.7, between the multiple biomarkers in septic shock patients compared to critical COVID-19 patients (Appendix A). Among circulating nucleosomes and neutrophil activation parameters, Nu.Cit-H3R8 and Cit.H3 were correlated (i.e., r Pearson > 0.7) in both septic shock and critical COVID-19 patients. Nevertheless, differences were observed between septic shock and critical COVID-19 patients. While in septic shock patients, Nu. H3.1 correlated (i.e., r Pearson > 0.7) with MPO and NE, these correlations were not observed (i.e., r Pearson < 0.2) in critical COVID-19 patients (Appendix A). Correlations between Nu.H3.1 and APACHE-II (r Pearson = 0.471, 95% CI: 0.209 to 0.669 and −0.432, 95% CI: -0.722 to -0.0128 for septic shock and critical COVID-19, respectively), SOFA (r Pearson = 0.609, 95% CI: 0.384 to 0.764 for septic shock) and NE (r Pearson = 0.719, 95% CI: 0.542 to 0.835 for septic shock) were significant (Appendix A).

## 4. Discussion

Viral infections encompass a broad spectrum of pathogens and diseases in humans but are rarely the primary cause of sepsis [21]. Historically, influenza has been one of the more common viral causes of sepsis. However, it is unclear to what extent the primary viral infection as opposed to bacterial pneumonia co-infection is the cause of organ dysfunctions in these patients. SARS-CoV-2, causing COVID-19, is now responsible for many cases of infection and sepsis and the exploration of the underlying physiopathological mechanisms of critical COVID-19 patients versus “traditional” septic shock deserves to be investigated. Although the initial aim of this study was not to directly compare critical COVID-19 patients with septic shock patients, our cohort permitted initial exploratory analyses, which permitted outlining the beginnings of more targeted investigations.

The results from this study confirm previous observations from other groups that COVID-19 and sepsis patients have different thrombo-inflammation profiles (Appendix A) [18,27,28]. Circulating nucleosomes and neutrophil activation markers were higher in septic shock and critical COVID-19 patients compared to the control group. Nevertheless, only levels of Nu.H3.1, a global marker of nucleosome release, and NE differ between critical COVID-19 and septic shock patients. While higher Nu.H3.1 levels are observed in critical COVID-19 compared to septic shock patients, an opposite trend is reported for NE and to a lesser extent for Nu.H3R8 (Figure 1). Although septic shock included more subjects with cancer and chronic kidney disease (CKD), the exclusion of these subjects did not change this conclusion, i.e., Nu.H3.1 titer is higher in critical COVID-19 patients and NE is higher in septic shock patients (data not shown).

As levels of NETs’ formation biomarkers have been reported to be linked with neutrophilia [10,29], we computed ratios of circulating nucleosomes and neutrophil activation markers according to the neutrophil count to estimate the distinctive contribution of neutrophils to the generation of circulating nucleosomes and the degree of neutrophil activation. We found that the ratio of Nu. H3.1/neutrophils was higher in critical COVID-19 patients compared to septic shock patients, a difference that is more pronounced than the difference of Nu.H3.1 alone (Figure 1 and Table 2). This highlights that Nu.H3.1 may also originate from other cell types in critical COVID-19 patients and that the contribution of these cell types could differ between critical COVID-19 and septic shock patients. In addition, NE, which was statistically higher in septic shock compared to critical COVID-19, became non-significant when divided by neutrophil counts (Figure 1). The level of Nu.H3R8/neutrophils did not differ between the groups even when stratified by clinical severity scores or when CKD and cancer patients are removed from the septic shock group. Nevertheless, we cannot determine the proportion of neutrophils that release NET in our patients and, although we can postulate that the ratio we computed is interesting, we have to admit that another hypothesis could be that a different proportion of neutrophils enter into a NET formation phase.

However, and interestingly, in septic shock patients, the levels of NE correlated with Nu.H3.1 (r Pearson = 0.790 in septic shock population versus 0.172 in COVID-19 population), suggesting that a higher proportion of circulating nucleosomes may originate from neutrophils compared to critical COVID-19 patients (Appendix A). Such observations permit us to reasonably hypothesize that circulating nucleosomes in septic shock are associated with NETs’ formation, while in critical COVID-19, it may originate from other cell types known to release chromatin fibers such as monocytes [30] and mast cells [31]. These cell types are also responsible for the pro-inflammatory state observed in COVID-19 [32,33], and this is consistent with the higher levels of IL-1β, IP-10 and IL-5 observed in the critical COVID-19 group compared to the septic shock group (Appendix A). Levels of Nu.Cit-H3R8, Cit-H3 and MPO, either normalized by neutrophil count or not, as well as NE/neutrophils, are not different in septic shock and critical COVID-19 cohorts, suggesting similar PAD4 activity in these groups. Thus, as PAD4 is mainly expressed by hematopoietic cells, with the highest levels in neutrophils [34], the absence of a difference suggests that the global citrullination activity is not different between the critical COVID-19 and septic shock groups, supporting the assumption that a part of the circulating Nu.H3.1 should originate from cell types other than neutrophils.

Levels of circulating nucleosomes and neutrophil activation markers were also evaluated according to APACHE-II and SOFA scores (Figure 3 and Figure 4). Interestingly, for similar APACHE-II and SOFA scores, Nu.H3.1 levels were higher in critical COVID-19 compared to septic shock patients, confirming that the difference observed in this biomarker between the two cohorts is probably not explicable by the overall pathological damage but results from different physiopathological processes. Interestingly, in septic shock patients, there is a correlation between Nu.H3.1 levels and APACHE-II and SOFA scores (Appendix A). The low sample size of the critical COVID-19 cohort (i.e., 16 patients with APACHE-II 0–15 and 6 patients with APACHE-II 16–25 and 20 patients with SOFA 0–6 and 2 patients with SOFA 7–9 scores, no patient with higher APACHE-II or SOFA scores) prevents such an analysis but correlations seem less clear. In septic shock, higher APACHE-II and SOFA scores were also associated with higher ratios of Nu.H3.1/neutrophils and Nu.Cit-H3R8/Nu.H3.1 (Table 2 and Figure 5). The other markers such as MPO and NE were not statistically associated with higher APACHE-II or SOFA scores, consistent with previous investigations [35]. Nevertheless, we can observe a trend for higher levels of these markers with clinical severity. This deserves to be further investigated.

These results are in line with the study of Cavalier et al. who found that higher levels of Nu.H3.1 were observed in COVID-19 patients compared to non-COVID hospitalized patients [12]. These authors also reported higher Nu.Cit-H3R8 levels in ICU patients compared to outpatients or patients in regular wards, an observation consistent with the difference we observed between our ICU patients and the control group [12]. Higher SOFA scores, i.e., ≥13, were also associated with higher MPO levels (Figure 4). Although the differences were not statistically significant, we also reported a trend towards higher levels of Nu.Cit-H3R8 and NE according to disease severity (Table 2, Figure 3 and Figure 4), consistent with previous observations on NETs’ formation markers relative to APACHE-II and SOFA scores [36].

In this cohort, we did not find an association between 30-day mortality or thrombotic events with levels of NETs’ formation markers (Figure 2), although a trend was observed for Nu.H3.1 (median [10th–90th percentile]: 1150 [223.8–4310] ng/mL versus 1199 [480.8–15,863] ng/mL, *p* = 0.0982).

NETs’ formation is a regulated process that is involved in both chronic and acute mechanisms, differing in their stimuli [37]. In addition to classical biomarkers of inflammation, NETs’ formation, markers of neutrophil activation and circulating nucleosomes represent additional and complementary markers to assess disease severity and global cell death in patients suffering from ARDS. Nucleosomes can be released from multiple cell types following cell death resulting from disease progression or multiple organ failure, as confirmed by our results [38]. Interestingly, citrullinated histones, either Nu.H3R8 or Cit-H3, do not differ between septic shock and critical COVID-19, suggesting that these two medical conditions trigger the citrullination of histones and nucleosomes similarly. Processed nucleosomes, circulating histones and citrullinated histones are important potential contributors of cytokine storm [39]. Although citrullination is dispensable to further initiate NETs’ formation, it potentiates histone-related signaling [13]. As depicted in this study, measuring both Nu.H3.1 and Nu. Cit-H3R8 could first permit us to identify the global disease severity by measuring circulating nucleosomes resulting from multiple cell death as represented by Nu.H3.1 levels. And, second, the use of the ratio of circulating Nu.Cit-H3R8 on Nu.H3.1 could further inform on the NETs’ formation. Thus, while cell death leads to increased nucleosome levels generally, the increase in citrullinated nucleosomes is consistent with a hyperinflammatory response associated with septic shock and critical COVID-19.

This study has limitations since the COVID-19 group was relatively small, so results must be validated in a larger confirmation cohort. However, the prospective and systematic enrollment of patients with either COVID-19 or septic shock within a closed period has limited inclusion bias. In addition, the observations were based on a single time point, namely, early after ICU admission, and a longitudinal assessment of these circulating nucleosomes and neutrophil activation biomarkers could better define the dynamic changes over time, which remain to be clarified in both septic shock and critical COVID-19 patients. Another point to highlight is that patients were not treated with similar therapies, and thus a specific “treatment effect” cannot be excluded to explain the differences observed. Finally, it will be important to assess whether these biomarkers may correctly discriminate between severe and non-severe patients and therefore predict poor outcomes at the individual level. This study could also pave the way to investigate these two clinical conditions with in vitro cellular models in order to delineate in more details the difference between these two NET formation processes. Standardization of the techniques used for the exploration of NETs’ formation is also mandatory as the literature is growing rapidly in this field. Nevertheless, these results are encouraging and may serve as hypotheses generating for the development of clinical decision algorithms. If at an individual patient level NETs’ formation is considered as the main trigger of cell damage and the global inflammatory process, this may serve for clinical decision making and permit the administration of targeted therapies. On the other hand, if this is not the main contributor, other therapies that instead target the inflammation processes could be administered. Thus, even if these results might seem limited in their application now, the understanding of different potential physiopathological processes might be valuable in the future.

## 5. Conclusions

This study reveals that Nu.H3.1 and Nu.Cit-H3R8 appear to be potential markers of global cell death and neutrophil activation when combined. Nu.H3.1 permits the evaluation of disease severity and differs between critical COVID-19 and septic shock patients, reflecting two distinct potential pathological processes in these ARDS conditions. The normalization of Nu.H3.1 on the neutrophil count permits us to better discriminate these different populations, reflecting the higher contribution of neutrophils to generate nucleosomes in septic shock patients. Nevertheless, the ratio of Nu.Cit-H3R8, Cit-H3, NE and MPO levels on neutrophils were similar between the two cohorts, suggesting a similar NETs’ formation potential in critical COVID-19 and septic shock patients admitted to ICU. Further studies are required to confirm if the measurement of nucleosomes and citrullinated nucleosomes may predict disease severity and help in categorizing patients at an early stage of the disease.

## Figures and Tables

**Figure 1 biomolecules-12-01038-f001:**
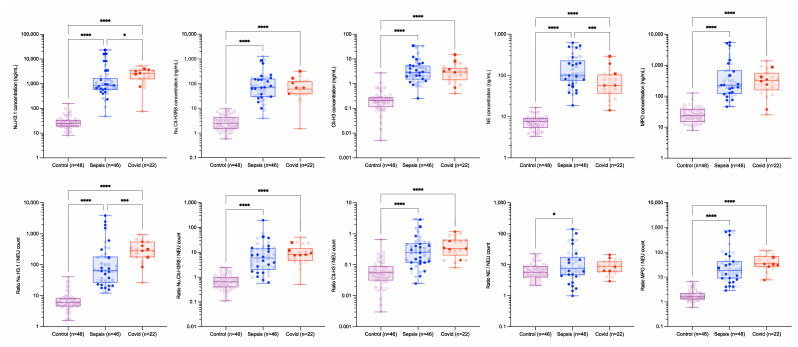
**Levels of circulating nucleosomes and neutrophil activation biomarkers in control, septic shock and critical COVID-19 populations.** Nu.H3.1, Nu.Cit-H3R8, Cit-H3, NE and MPO were compared. Results are expressed as absolute value or normalized by neutrophils level for each individual. All markers were statistically different in septic shock and critical COVID-19 compared to controls. Only Nu.H3.1 and NE were different between septic shock and critical COVID-19 patients. Boxes represent 25th–75th percentile with median. Whiskers represent min to max variation. Squares represent patients with a thromboembolic event, and non-transparent symbols represent dead patients. *, *** and **** represent *p*-values ≤ 0.05, ≤0.001 and ≤0.0001, respectively. Only differences that are statistically significant are reported. Abbreviations: Cit-H3, citrullinated histone H3 (citrullinated in R2, R8 and R17); MPO, myeloperoxidase; NE, neutrophil elastase; Nu.Cit-H3R8, citrullinated H3R8-nucleosome; Nu.H3.1, H3.1-nucleosome.

**Figure 2 biomolecules-12-01038-f002:**
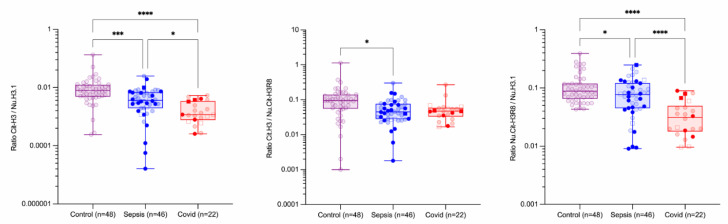
**Ratio of circulating nucleosomes and neutrophil activation parameters in control, septic shock and critical COVID-19 subjects.** The following ratios were proposed: (i) ratio of Cit-H3 on Nu.H3.1, (ii) ratio of Cit-H3 (citrullinated in R2, R8 and R17) on Nu.Cit-H3R8 and (iii) ratio of Nu.Cit-H3R8 on Nu.H3.1. Boxes represent 25th–75th percentile with median. Whiskers represent min to max variation. Squares represent patients with a thromboembolic event, and non-transparent symbols represent dead patients. *, *** and **** represent *p*-values ≤ 0.05, ≤0.001 and ≤0.0001, respectively. Only differences that are statistically significant are reported. Abbreviations: Cit-H3, citrullinated histone H3 (citrullinated in R2, R8 and R17); Nu.Cit-H3R8, citrullinated H3R8-nucleosome; Nu.H3.1, H3.1-nucleosome.

**Figure 3 biomolecules-12-01038-f003:**
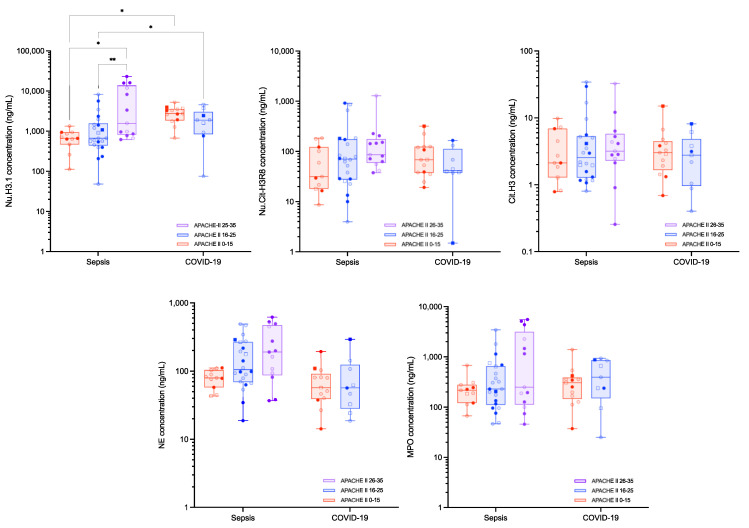
**Circulating nucleosomes and neutrophil activation parameters in septic shock and critical COVID-19 subjects according to the APACHE-II score**. Boxes represent 25th–75th percentile with median. Whiskers represent min to max variation. Squares represent patients with a thromboembolic event, and non-transparent symbols represent dead patients. *, ** represent *p*-values ≤ 0.05 and ≤0.01, respectively. Only differences that are statistically significant are reported. Abbreviations: Cit-H3, citrullinated histone H3 (citrullinated in R2, R8 and R17); MPO, myeloperoxidase; NE, neutrophil elastase; Nu.Cit-H3R8, citrullinated H3R8-nucleosome; Nu.H3.1, H3.1-nucleosome.

**Figure 4 biomolecules-12-01038-f004:**
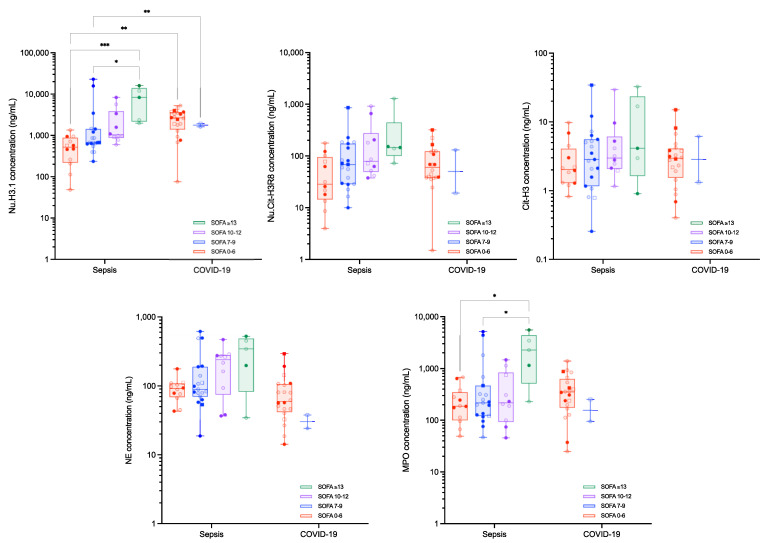
**Circulating nucleosomes and neutrophil activation parameters in sepsis and COVID-19 subjects according to the SOFA score.** Boxes represent 25th–75th percentile with median. Whiskers represent min to max variation. Squares represent patients with a thromboembolic event, and non-transparent symbols represent dead patients. *, ** and *** represent *p*-values ≤ 0.05, ≤0.01 and ≤0.001, respectively. Only differences that are statistically significant are reported. Abbreviations: Cit-H3, citrullinated histone H3 (citrullinated in R2, R8 and R17); MPO, myeloperoxidase; NE, neutrophil elastase; Nu.Cit-H3R8, citrullinated H3R8-nucleosome; Nu.H3.1, H3.1-nucleosomes.

**Figure 5 biomolecules-12-01038-f005:**
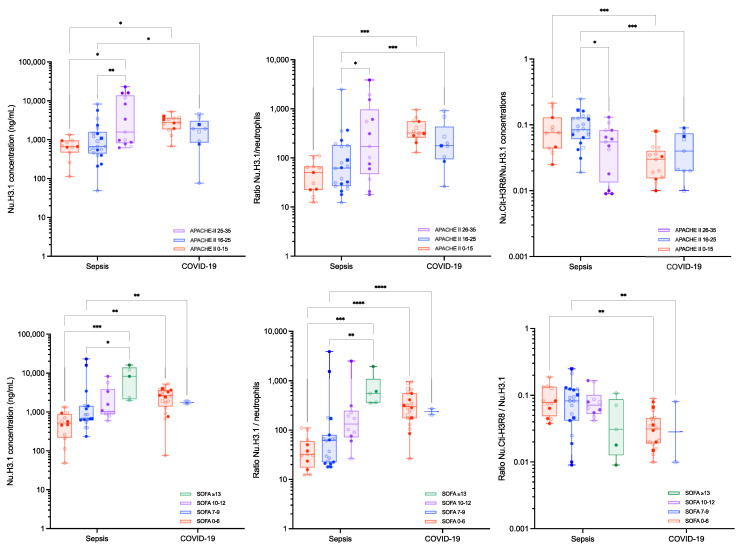
**Comparison of nucleosome markers and derived ratio calculation according to APACHE-II and SOFA scores in septic shock and critical COVID-19 subjects.** Boxes represent 25th–75th percentile with median. Whiskers represent min to max variation. Squares represent patients with a thromboembolic event, and non-transparent symbols represent dead patients. *, **, *** and **** represent *p*-values ≤ 0.05, ≤0.01, ≤0.001 and ≤0.0001, respectively. Only differences that are statistically significant are reported. Abbreviations: Cit-H3, citrullinated histone H3 (citrullinated in R2, R8 and R17); MPO, myeloperoxidase; NE, neutrophil elastase; Nu.Cit-H3R8, citrullinated H3R8-nucleosome; Nu.H3.1, H3.1-nucleosome.

**Table 1 biomolecules-12-01038-t001:** **Comparison of NETs’ formation markers in survivors and non-survivors.** Results were reported as median (10th–90th percentile). Differences between groups were assessed using an unpaired *t*-test on log transformed data.

		Nu.H3.1 (ng/mL)	Nu.H3R8 (ng/mL)	Cit.H3 (ng/mL)	NE (ng/mL)	MPO (ng/mL)

All cohort						
Survivors (n = 42) Median (10th–90th percentile)		1081.4 [222.3–4295.4]	54.3 [14.3–184.9]	2.6 [0.8–8.9]	86.7 [33.3–285.1]	251.8 [74.5–1312.2]
Non-survivors (n = 26) Median (10th–90th percentile)		1778.3 [439.5–15,848.9]	76.0 [23.1–396.3]	3.1 [0.8–12.9]	96.6 [30.1–504.7]	235.0 [64.1–4581.4]
*p*-value		0.0982	0.1178	0.4818	0.3573	0.7197
** Septic shock **						
Alive (n = 26) Median (10th–90th percentile)		785.2 [173.4–3076.1]	65.5 [11.7–299.2]	2.6 [0.8–20.0]	106.2 [59.4–374.1]	239.9 [61.1–1932.0]
Death (n = 20) Median (10th–90th percentile)		901.6 [402.7–16,032.5]	76.0 [18.7–592.9]	2.9 [0.9–11.8]	96.6 [37.0–523.6]	210.4 [74.3–5046.6]
*p*-value		0.0664	0.4007	0.9598	0.8881	0.6946
** COVID-19 **						
Alive (n = 16) Median (10th–90th percentile)		1927.5 [353.2–4764.3]	41.9 [8.9–154.2]	2.5 [0.7–6.3]	48.9 [22.5–117.5]	301.3 [63.8–1052.0]
Death (n = 6) Median (10th–90th percentile)		2964.8 [765.6–3999.4]	85.9 [39.1–318.4]	3.5 [0.7–15.0]	79.3 [14.3–292.4]	326.6 [37.3–875.0]
*p*-value		0.6946	0.1442	0.2272	0.2349	0.9187

**Table 2 biomolecules-12-01038-t002:** Circulating nucleosome and histone parameters in septic shock and critical COVID-19 patients according to APACHE-II and SOFA scores. Results are reported as median with the 10th–90th percentile. Data are compared using an ordinary two-way ANOVA with *p*-value corrected for multiple comparisons using a Tukey’s multiple comparisons test.

		APACHE-II 0–15		APACHE-II 16–25	APACHE-II 25–35		SOFA 0–6	SOFA 7–9	SOFA 10–12		SOFA ≥ 13	

** Nu.H3.1 (ng/mL) **												
**Septic shock**		666.4 (133.7–1257.8)		670.0 (215.9–4898.9)	1575.3 (641.4–19,955.7)		517.9 (62.6–1213.9)	673.0 (396.9–15,775.5)	1032.8 (612.7–7993.8)		8285.6 (1980.4–16,068.7)	
**Critical COVID-19**		2764.5 (877.8–4720.9)		1904.0 (76.0–4555.3)			2648.3 (689.2–4496.4)	1769.1 (1615.1–1937.8)				

corrected *p*-value		0.0321	0.0321				0.0025	0.025				

** Nu.H3.1/neutrophils **												
**Septic shock**		50.5 (13.2–108.9)		62.3 (18.8–362.7)	172.4 (19.0–2934.4)		31.8 (12.4–107.9)	61.4 (18.0–1533.1)	132.7 (28.7–2025.4)		553.7 (353.0–1933.7)	
**Critical COVID-19**		322.6 (155.5–826.3)		177.4 (26.4–914.7)			299.5 (86.9–888.9)	236.0 (204.4–272.4)				

corrected *p*-value		0.0005	0.0005				<0.0001	<0.0001				

** Nu.Cit-H3R8 (ng/mL) **												
**Septic shock**		31.6 (10.2–183.6)		70.2 (11.0–798.3)	86.2 (39.3–862.1)		28.6 (5.4–161.1)	68.5 (16.5–226.3)	79.1 (38.0–893.3)		145.7 (72.3–1286.0)	
**Critical COVID-19**		68.4 (21.9–271.5)		41.1 (1.5–165.9)			60.6 (25.6–218.6)	75.3 (19.2–131.4)				

corrected *p*-value		>0.9999	>0.9999				0.9538	0.9538				

Cit-H3 (ng/mL)												
**Septic shock**		2.12 (0.79–9.28)		2.58 (1.1–25.8)	3.18 (0.52–24.5)		2.04 (0.94–8.92)	2.84 (0.79–12.1)	2.99 (1.24–27.6)		4.14 (0.91–32.7)	
**Critical COVID-19**		3.04 (0.94–11.7)		2.77 (0.40–2.77)			2.96 (0.71–8.01)	3.74 (1.32–6.15)				

corrected *p*-value		0.9966	0.9966				>0.9999	>0.9999				

Nu.Cit-H3R8/Nu.H3.1												
**Septic shock**		0.076 (0.027–0.209)		0.084 (0.034–0.165)	0.055 (0.009–0.121)		0.079 (0.040–0.172)	0.083 (0.010–0.214)	0.072 (0.043–0.166)		0.031 (0.009–0.108)	
**Critical COVID-19**		0.030 (0.011–0.064)		0.040 (0.010–0.090)			0.031 (0.013–0.079)	0.028 (0.010–0.081)				

corrected *p*-value		0.0002	0.0002				0.0038	0.038				


Abbreviations: Cit-H3, citrullinated histone H3; MPO, myeloperoxidase; NE, neutrophil elastase; Nu.Cit-H3R8, citrullinated nucleosome H3R8; Nu.H3.1, nucleosome H3.

## Data Availability

The raw data supporting the conclusions of this article will be made available by the authors, without undue reservations.

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
