# Peer review of "NETosis and Nucleosome Biomarkers in Septic Shock and Critical COVID-19 Patients: An Observational Study"

_biomolecules, 2022, doi:10.3390/biom12081038_

Round 1

Reviewer 1 Report

The manuscript of Morimont et al reports on the discriminative capacity of NETosis and nucleosome biomarkers in septic shock and critical COVID-19 patients. The results are interesting as the study includes several novel biomarkers assessed in the plasma of patients with different pathologies caused by bacterial and viral infections. These conditions are potentially associated with different kinds of reactions of the body toward the pathogens. Therefore, the authors included markers like citrullinated nucleosomes, neutrophil elastase (NE) and myeloperoxidase (MPO) known to be involved in neutrophil activation and NETosis, as well as more non-specific markers like H3.1 nucleosomes. While all markers were elevated in both diseased groups as compared with controls, differences between patients with septic shock and COVID-19 were only observed for H3.1 and NE. In addition, various ratios were calculated to increase the discriminative power. Correlations with disease severity were only observed for H3.1 in sepsis but not COVID-19, and no association with 30 day survival was observed. The set of markers has been further enriched by plenty of cytokines and chemokines to show the different inflammatory pattern between the two pathologies.

While the study addresses a novel and pathophysiologically attractive question, some major concerns have to be addressed:

1.    As outlined by the authors in the discussion section, the number of patients in each group is rather small lending the study only an explorative character. This is supported by the lack of correction to multiple testing and testing many types of biomarker ratios. It is clear that further subgroup and more detailed analyses are limited and can only give preliminary results that have to be confirmed in validation studies.

2.    It is unclear whether the study question can really be answered by the study design. Severely diseased patients obviously have increased levels of multiple biomarkers, however, the discrimination of bacterial and viral infections is hardly possible by these markers. Several cytokine markers showed greater differences. However, it is not discussed what would be the clinical value of discriminating the two distinct patient cohorts? What would be the benefit of knowing the sensitivities and specificities for this discrimination? Unfortunately there is also no clear association with disease severity and survival.

3.    It is not clear why NETosis markers should discriminate between the pathologic conditions if NETosis is linked to both of them. If different kinds or proportions of NETosis play a role, then the results have to be discussed more critically: Citrullination obviously is not relevant neither MPO. H3.1 is higher in COVID-19, NE and neutrophils in septic shock. It is self-evident that the formation of appropriate ratios shows the respective differences, too (e.g. H3.1/neutros or H3R8cit/H3.1). The discussion seems to be quite speculative given the thin basis of data, the only one-time measurements and the clinical and therapeutic heterogeneity within the groups. On the other side, other aspects like the higher numbers of CKD and cancers in the septic shock group are neglected.

4.    If different types of NETosis (like vital and suicidal) are involved, the relevance of the release of the named markers should be presented or at least discussed. Are results from cell model studies available? How strong is the dependency on the pathogen exposed? Do antibiotics or other treatment influence these processes? Is something known about the half-time and dependency of marker concentrations from local degradation, metabolization and elimination? Otherwise hypotheses deducted from these results have to be reduced or put into perspective.

Minor points:

5.    As the assays are new, data on the analytical performance and quality control should be given.

6.    In Fig 3a, the bar for Apache 26-35 is missing

7.    In Tab 3, numbers have to be checked. E.g. for H3R8/H3.1, septic shock, Apache II, the median is outside the range. In addition, adjusted p-values should be explained.

8.    Results (lines 280-284) have to be checked. Correlations with disease severity are different from the graphical presentation in Fig3 and 4.

9.    Fig 5 should be placed into the results section

10. Calculations for the correlation with survival should be added.

Reviewer 2 Report

The presented study provides some new insight on the NETs release in COVID and critically ill patients, but I believe that readers could benefit more from the manuscript, had it been provided in a more concise, easy-to-follow manner. Below you can find my specific comments.

Abstract

1. Line 26 - I believe it should be written what are the exact differences that have been found

2. It is written, as a result, that:

The ratio of Nu.Cit-H3R8 to Nu.H3.1 allows the determination of nucleosome citrullination degree, presumably by PAD4, reflecting the involvement of NETosis in the total amount of nucleosomes generated

the correlation of citH3 and NETs release has not been proven in the study and should not be described as a result

Introduction

1. Using the term "NETosis" as a synonym for NETs release is erroneous and not in the line with the current knowledge in the field. NETs release, including elevated content of the nucleosomes, does not necessarily result from cell death (NETosis). Please follow the current recommendations regarding nomenclature (https://www.nature.com/articles/s41418-018-0261-x) and avoid using the term NETosis.

2. Lines 64-65 - antimicrobal activity of which cells is reduced? It is not clear whether this passage refers to neutrophils or monocytes.

3. Lines 87 - 89 It is unclear in which conditions the levels of various markers, eg pPLT counts, are higher - in COVID or sepsis?

4. The aim of the study is not clearly presented. Actually, the last line of the introduction only states which studies were not part of the author's intial study.

Materials and methods

1. It seems dubious, why oncologic patients constitute such a high fraction of control group while COVID group contains no such patients. Similarly with smokers. If the fraction of smoking/oncologic patients was not kept similar in the control group as well as in study groups, the reasons for this should be listed in the manuscript. 

2. Regarding inclusion criteria - the minimum time of 1 month from the last chemotherapy or 48 hours after major surgery seem to be rather short. Can the authors explain why such not very strict criteria were applied?

3. Since Table 1 is actually a repetition of previously publihed data, I would find it more appropriate to move it to the supplement. 

4. Table 1

It should be written in the table description what comparisons/statistical tests have been made, for which p values are presented.

In some cells of the table it is not clear what values are shown there - means with SD? Medians with SEM? Please add this information.

From the first column it can be seen that e.g. the number of men will be presented as follows: (n,%), which implies that both numbers will be provided in brackets. Please note that only % is written in brackets. Accordingly, it should be written: n (%)

Results

1. Is the table 2 a repetition of previously published data? If so, it should be moved to a supplement. I can guess why some parameters are in bold but it should be clearly stated in the description.

2. Lines 199-200 - check English, in the second part of a sentence a verb is lacking.

3. Overall, I find this section overloaded with numbers and therefore not very easy to follow. If the autors feel like exact p values, medians, 10-90 percentile etc are important, they can be easily provided in tables but the main text would be much more straightforward and "smooth" without these details.

4. Regarding p values, it is written that e.g. * is p<0.05, whereas Prism uses less-than-or-EQUAL (<=0.05), please refer to: https://www.graphpad.com/support/faq/what-is-the-meaning-of--or--or--in-reports-of-statistical-significance-from-prism-or-instat/

and correct mathematical symbols throughout the manuscript. 

5. Lines 248-249 - it seems unclear what do the authors mean by saying that the ratios did not improve stratification. What was cheked/compared to assess the potential improvement of stratification? Please clarify.

6. Table 3 - it should be clearly stated in the description, to what type of comparisons do the p-value refer to. As I understand, COVID vs sepsis comparisons have been made, and only patients with scores 0-25 for APACHE or 0-9 for SOFA were included? 

7. Figure 3 - why the group of septic patients with APACHE score of 26-35 is not included in the first of the sub-images (Nu.H3.1  concentration)?

8. Lines 252-253. It is written that:

Nu.H3.1 levels and the ratio of Nu.H3.1/neutrophils were superior in critical COVID-19 patients versus septic shock patients while the ratio of Nu.Cit-H3R8/Nu.H3.1 was lower in COVID-19 patients (Figure 4 and Table 3).

According to the figure 4, the highest values of Nu.H3.1 were observed in the group of septic patients, which is contrary to the aforementioned sentence. According to the table 3, septic shock patients have the median value of 8285 when the SOFA score is at least 13. It is far more than in critical COVID patients. Please explain. 

9. Although no statistical significance was observed, multiple trends in data presented in Fig. 3 and 4 are seen. In my opinion, the authors should comment these changes.

10. Supplementary Figure 2: in the description, it is written that both positive and negative strong correlations (>0.7, <-0.7) are marked in bold. However, according to the captions on the right side of each of three panels, the list contains only positive correlations. Which is true?

11. "There was a stronger correlation, as defined by a Pearson's r above 0.7, between the 273 multiple biomarkers in critical COVID-19 patients compared to septic shock patients" - it is unclear how this conclusion is drawn. Were the r values, on average, higher in the group of COVID patients? The number of correlations above 0.7 in bold is the same in both groups (8 pairs are in bold), so I understand it is not what you meant by this sentence. Please clarify.

12.What does it mean that  in septic shock patients, Nu. H3.1 correlated WELL with MPO and NE? These correlations are not in bold, so thay are below 0.7.

13. Data from supplementary figure 3 are described twice in a similar manner - please note lines- 240-243 and 281-284. Please delete one of these descriptions.

Discussion

1. I do not think that computing different indexes according to the neutrophil count allows primarily to estimate the distinctive contribution of neutrophils to the generation of circulating nucleosomes. I see it as a way to, most of all, exclude that observed differences in the concentration of NE and nucleosomes are due to various numbers of circulating granulocytes.

2. It is sometimes difficult to follow what is the reasoning of the authors, e.g. in paragraph 2 of the discussion.

3. In some instances, no data are available to back up the hypotheses. Accordingly, the sentences such as the following:

…highlighting that Nu.H3.1 originates from different cell types and that the contribution of these different cell types differs between critical COVID-and septic shock.

should be rephrased into

“may originate from different cell types”

4. The authors argue that similar concentrations of citrullinated histones in both groups (COVID and sepsis) suggest similar PAD4 activity. Yet, the total concentration of nucleosomes is higher in covid patients, which means that the fraction of citrullinated histones among total histones is lower. Accordingly, it would suggest that the activity of PAD4 is lower in this subgroup – the enzyme had high chances to encounter its substrate, yet the amount of citrullinated protein is only at the same level as in sepsis group. The authors should refer to this issue.

Conclusions

4. The first sentence of conclusion section is rather vague. What does it mean, that markers are interesting? Please try to express this conclusion more precisely.

Round 2

Reviewer 2 Report

I think that the manuscript overall improved and should be accepted for publication after some minor, remaining issues are corrected.

1.        

Reviewer, round 1: Supplementary Figure 2: in the description, it is written that both positive and negative strong correlations (>0.7, <-0.7) are marked in bold. However, according to the captions on the right side of each of three panels, the list contains only positive correlations. Which is true?

Response to the reviewer:

All correlations that are reported next to the correlation matrix are correlation with Pearson’s r above 0.7. We found no negative correlation (i.e., <-0.7). As reported in the legend, parameters showing a correlation above 0.7 (or below – 0.7) in both septic shock and critical COVID-19 cohorts are marked in bold.

Reviewer, round 2: Ok, I understand now what the bolding means, but I find the caption slightly misleading. I would suggest rephrasing into:

Supplementary Figure 2: Correlation matrix of all parameters reported in this study in controls, septic shock and critical COVID-19 cohorts. Parameters showing a correlation above 0.7 in both cohorts (septic shock and critical COVID-19 patients) are marked in bold. No correlations below – 0.7 were found.

2.        

Reviewer, round 1 : what does it mean that in septic shock patients, Nu. H3.1 correlated WELL with MPO and NE? These correlations are not in bold, so they are below 0.7.

Response to the reviewer:

The correlation of Nu.H3.1 and MPO and NE is above 0.7 as reported in the table.

Reviewer, round 2: My question was about wording, the word “well”  is rather vague in the context of correlation. We simply say that the correlation is strong if r is (>0.7), or it is positive or negative. The word WELL should be removed or replaced by an appropriate term.

3.       There is no point for the new sentence put in line 282, since it means exactly the same as the sentence in line 281

4.       Line 285 “such observations permits” – the verb should be plural.

5.       Lines 287-288 read: “(…) while in critical COVID-19, it may originate or from other cell types known to release chromatin fibers like monocytes and mast cells.

“OR” is probably put there as a result of a mistake

6.       Line 354 – it should be literature instead if litterature
